# The Transcriptome of Chicken Migratory Primordial Germ Cells Reveals Intrinsic Sex Differences and Expression of Hallmark Germ Cell Genes

**DOI:** 10.3390/cells12081151

**Published:** 2023-04-13

**Authors:** Dadakhalandar Doddamani, Mark Woodcock, Lorna Taylor, Sunil Nandi, Lynn McTeir, Megan G. Davey, Jacqueline Smith, Mike J. McGrew

**Affiliations:** The Roslin Institute and Royal (Dick) School of Veterinary Studies, University of Edinburgh, Easter Bush Campus, Edinburgh EH25 9RG, UK

**Keywords:** PGC, transcriptome, stem cell, gametogenesis, sex determination, chicken

## Abstract

Primordial germ cells (PGCs) are germline-restricted embryonic cells that form the functional gametes of the adult animal. The use of avian PGCs in biobanking and producing genetically modified birds has driven research on the in vitro propagation and manipulation of these embryonic cells. In avian species, PGCs are hypothesized to be sexually undetermined at an early embryonic stage and undergo differentiation into an oocyte or spermatogonial fate dictated by extrinsic factors present in the gonad. However, chicken male and female PGCs require different culture conditions, suggesting that there are sex-specific differences, even at early stages. To understand potential differences between male and female chicken PGCs during migratory stages, we studied the transcriptomes of circulatory stage male and female PGCs propagated in a serum-free medium. We found that in vitro cultured PGCs were transcriptionally similar to their in ovo counterparts, with differences in cell proliferation pathways. Our analysis also revealed sex-specific transcriptome differences between male and female cultured PGCs, with notable differences in Smad7 and NCAM2 expression. A comparison of chicken PGCs with pluripotent and somatic cell types identified a set of genes that are exclusive to germ cells, enriched in the germplasm, and associated with germ cell development.

## 1. Introduction

Embryonic primordial germ cells (PGCs) are central to genetic inheritance as they give rise to the functional gametes of an animal. The use of PGCs for genetic modification and the potential production of in vitro gametes has driven research on their in vitro culture and the identification of sex-specific differences [1,2]. Cryopreserving avian germplasm is difficult due to the yolk filled egg and the low fertility obtained in many bird species from frozen avian sperm. PGCs offer an alternative cell type for the biobanking of bird species [3]. In chickens, the laid egg stage embryo consists of 50,000–60,000 blastodermal cells, including approximately 50–60 PGCs, which proliferate during their migration towards the genital ridges through the vascular system [4]. After colonizing the genital ridges at stage HH16 (day 2.5), both male and female PGCs proliferate until stage HH34 (day 8) and subsequently follow different differentiation pathways in male and female embryos, leading to the formation of functional sperm and oocytes, respectively [5].

In birds, it was shown that incomplete dosage compensation of the Z chromosome led male ZZ somatic cells to have a distinct cellular transcriptional identity compared to female ZW somatic cells at all stages of development [6]. This is particularly evident in reproductive tissues, where male and female somatic cells differ in their capacity to form male or female gonadal cell types, leading to the hypothesis of cell-autonomous sex identity (CASI) in avian species [7]. Avian germ cells were also hypothesized to be sexually determined for a male or female fate. The rate of germline transmission is higher when chicken PGCs are transplanted into same-sex host embryos in comparison to opposite-sex host embryos [8,9,10]. Previous studies also showed that in vitro propagated PGCs do not form functional gametes in opposite-sex hosts [11,12]. Supporting this idea, a proteomics analysis showed that male and female cultured PGCs have a distinct proteome [13]. Contrary to these results, it was recently shown that both in vitro propagated male and female chicken PGCs can form functional gametes in sterile opposite-sex hosts of both sexes [14,15], indicating that ZZ and ZW germ cells are both capable of completing spermatogenesis or oogenesis. These results led us to propose that circulatory male and female chicken PGCs are genetically distinct from each other, but the somatic cellular environment can override their genetic programming.

Recently, the transcriptomes of male and female PGCs isolated from blood at different embryonic stages showed transcriptome divergence [16]. To date, the germ cell-specific transcriptome of in vitro propagated chicken PGCs in a serum-free medium and the sexual differences in the gene regulatory networks at this stage for male and female PGCs have not been described. Furthermore, the disparities between cultured PGCs and freshly isolated PGCs have not been illustrated. Here, we characterize the differential gene regulatory networks in and between male and female chicken PGCs propagated in serum free medium and their in ovo counterparts. Taken together, our data suggest that early stages of development are cell-autonomous with distinct gene regulatory networks, and later stages are gonad-driven and regulate the formation of male and female gametes in birds.

## 2. Materials and Methods

### 2.1. PGC Culture Medium

Avian PGC culture medium contained B-27 supplement, 2.0 mM GlutaMax, NEAA, 0.1 mM β-mercaptoethanol, nucleosides, 1.2 mM pyruvate, 0.2% ovalbumin (Sigma), 0.2% sodium heparin in avian DMEM, and a custom basal medium (a modification of knockout DMEM (250 mosmol/L, 12.0 mM glucose, and CaCl-free)) [17]. The following growth factors were added before use: human Activin A, 25 ng/mL (Peprotech); human FGF2, 4 ng/mL; 0.2% ovotransferrin.

### 2.2. Derivation and Culturing of Chicken PGCs

Fertile Vantress broiler eggs (a kind gift from Cobb-Vantress Europe) were incubated for 55–60 h (stage 16 to 16^+^ H&H), followed by the isolation of blood (2.0 μL) from the embryo, which was added to a 300 μL FAOT culture medium [17]. Approximately 100 μL of the medium were changed on a routine basis every two days until the number of cells reached around 1 × 10^5^. At subsequent stages, the complete medium was changed at the same intervals until cells propagated to 4 × 10^5^ cells/mL medium. Cultured PGCs were frozen in Avian DMEM containing 4% DMSO/5% chicken serum and stored at −150 °C. The sex of the donor embryo was determined by isolating the embryo, as mentioned below. PGCs reached populations > 1 × 10^6^ cells and were assayed between 70 and 90 days in culture. Cells were collected from four male (5M, 19M, 29M, 57M) and five female (13F, 27F, 70F, 81F, 90F) individual cultures for RNA isolation, and total RNA was used for RNA-seq analysis. Control male and female PGC cultures from single eggs from Hy-line Brown layer chickens were used for RT-qPCR controls.

### 2.3. RNA Isolation and Quantification

Isolation of RNA from approximately 1 × 10^6^ cultured PGCs was conducted using the RNeasy Plus Micro Kit (Qiagen). The Agilent RNA tape station 2200 instrument was used to quantify the RNA. Samples with RIN values > 7.0 were considered suitable for sequencing. The library preparation and sequencing were carried out at the Edinburgh Genomics sequencing facility (Edinburgh). Illumina paired-end reads of length 150 base pairs were generated.

### 2.4. Generation of RNA-Seq Data and Genome Mapping

An average of ~119 million Illumina paired-end reads of 150 base pairs were obtained for each PGC line. These reads were subjected to quality control using NGS-QCbox [18], which reported Phred scores of ~96% (Q20) and ~92% (Q30). Adapter sequences were removed using the ‘Trimmomatic’ tool [19], retaining sequences that were longer than 75 bases. The trimmed reads were mapped against the chicken reference genome (GRCg6a) by using the HISAT2 aligner with default parameters [20].

### 2.5. Transcriptome Comparison with Other Cell Types

RNA-seq data from many different cell types was obtained from public databases and used to identify the genes that are exclusively expressed in chicken PGCs. The details of the datasets used in this analysis and their accession numbers are listed in Appendix A. The featurecounts tool (v2.0.2) [21] was used to count the number of reads mapped to genes by specifying -O (multioverlap), -P (paired end), and—primary options. The DESEQ2 R [22] package was used to compare the transcriptome data. To identify germ cell-exclusive genes, the absolute log2fold change value higher than three, an adjusted *p*-value lower than 0.05, and an average normalized expression value (obtained from DESEQ2) higher than 700 were considered the minimum gene expression values needed to be considered as a ‘positive’ expression in that particular group, and gene expression values less than 700 were taken as not being expressed. In addition, the ratio of expression values of PGCs to the expression values of non-germ cells being higher than 10 was also considered a qualifying criterion. PGC-specific genes were thus determined with the following parameters:

|log2fc| > 3, p.adj < 0.05, PGC_exp > 700, non-germ_exp < 700 and (PGC_exp/non-germ_exp) ≥10.

Gene ontology enrichment analysis was performed using germ cell-specific genes as an input into the Panther database [23]. The statistical over-representation test was selected for the GO complete biological process, and Fischer’s exact test and the false discovery rate correction were used for the FDR calculation. Protein–protein Interaction Network was constructed using the STRING database v11 [24], and the pathway analysis was performed using Ingenuity Pathway Analysis (IPA) software [25].

### 2.6. Transcriptome Comparison with In Ovo PGCs (E2.5)

RNA-seq data from Ichikawa et al. 2022 [16] was downloaded from the NCBI SRA database and followed the same analysis pipeline as mentioned above. We used DESEQ2 for differential expression analysis and used the following criteria for identifying differentially expressed genes between cultured and in ovo PGCs:

|log2fc| > 3, p.adj < 0.05, Avg Exp > 700.

Further, the gene enrichment analysis was performed by the Panther database.

### 2.7. RT-PCR to Validate Germ Cell Expression

A 15 μL total volume reaction mixture for PCR was made of 2 μL 1:10 diluted sample cDNA, 0.3 μL 10 mM dNTPs (Invitrogen), 0.3 μL 50 pmol/μL primers (i, ii), 1.5 μL 10X buffer + MgCl_2_, 0.1 μL Fast Start Taq, and 10.5 μL H_2_O. The reaction was carried out at 94 °C for 5 min, followed by 35 cycles of 94 °C for 30 s, 50 °C for 30 s, 72 °C for one min, and a final extension of 72 °C for 5 min. Samples were run on a 1% TAE agarose gel.

Samples used for the study were cDNA from chicken embryonic fibroblasts (CEF), the chicken embryo of day 6 (eD6) without mesonephroi as a positive control for somatic cell expression and a negative control for germ cell expression, chicken primordial germ cells (PGC) (Hy-line female sample), and the chicken day six embryo without the reverse transcriptase enzyme eD6 (-RT). The RT-PCR analysis for a housekeeping gene, *GAPDH,* confirmed the absence of genomic contamination in samples (Appendix A). The primers used for validation are listed in Appendix A. All these PGC-specific bands were sequenced for the confirmation of the correct product.

### 2.8. Preparation of Chicken Embryonic Fibroblasts (CEFs)

Fertile eggs were incubated for nine days, and embryos were dissected in a sterile ventilation hood and decapitated. In a sterile petri dish, the visceral material was removed from the embryos, and the remaining embryonic tissue was macerated using scissors. The embryos were then passed through a 10 mL syringe to partially homogenize the tissues, placed into 15 mL Falcon tubes, and 4 mL of trypsin-EDTA (Sigma-Aldrich, St. Louis, MO, USA) was added. Homogenized embryos were left to incubate at 37 °C for 15 min. An equal volume of STO media was added to halt the trypsin reaction, and the dissociated mixture was passed through a falcon cell strainer (100 μm). Cells were briefly centrifuged (at 5000 RPM for three minutes), and the supernatant was discarded and resuspended in fresh STO media. Cells were plated into T75 flasks and incubated at 37 °C.

### 2.9. In Situ Hybridization and Probe Synthesis

Probes, RNA polymerases, and restriction enzymes used are listed in Appendix A. In situ hybridizations were performed following the protocol in Henrique et al. [26].

### 2.10. Differential Gene Expression

The transcript annotation file from Ensembl release 96 (GRCg6a) was used during read counting using the FeatureCount tool. The matrix of raw read counts per gene was used in the DESeq2 R package to perform the normalization of the log transformation of read count data to the size of the library. The criteria of an average normalized read count higher than 500 FPKM was used to define genes expressed in PGC lines. The average expression value for a gene below 500 is considered a gene that is not expressed. DEGs were deemed significant if they had an absolute log2fold change value > 1 and an adjusted *p*-value < 0.05. The expression value of 500 was used as an additional criterion to remove false-positive DEGs.

Male PGCs: log2fc > 1, p.adj < 0.05 and male PGC_exp > 500.

Female PGCs: log2fc < −1, p.adj < 0.05 and female PGCs_exp > 500.

For dosage compensation, we used the log2fold change values of each gene between male and female PGCs. The median values of log2fold change for each chromosome are plotted using the ggplot2 package. The cumulative density of log2fold change of all autosomes and the Z chromosome was calculated and plotted using R scripts. 

### 2.11. Quantitative Reverse Transcription PCR (RT-qPCR)

RT-qPCR was performed to validate differentially expressed genes between male and female PGCs using primers listed in Appendix A. Reactions were set up in 96-well PCR plates and ran on an Mx300P thermal cycler. Setup and analysis were performed using MxPro software (Stratagene). We optimized RT-qPCR experiments by setting up reactions to amplify using different amounts of the same cDNA sample and picking primers that gave an R2 value higher than 0.95 and efficiency higher than 90%. Relative gene expression was calculated using the 2^−∆∆Ct^ method as described previously [27]. The GAPDH gene was used as a housekeeping gene for the RT-qPCR experiment. The −∆Ct value of a gene in a male sample was used as a reference to check the expression in female PGCs.

## 3. Results

### 3.1. RNA Transcriptome Analysis of Cultured PGCs

Chicken PGCs were isolated from single heritage line broiler embryos and cultured as stated in the Materials and Methods. An average of ~110 million trimmed reads were mapped to the chicken genome (GRCg6a). In total, an average of 92% of the reads could be mapped, and 67–81% of the reads could be mapped uniquely to a single locus in the genome (Appendix A). To confirm the authenticity of the generated data in this study, we first examined the RNA-seq data for the expression of *DAZL* and *DDX4,* which are known to have germ cell-specific expression. This analysis showed that, as expected, these two genes were highly expressed in the propagated PGCs (Appendix A). Furthermore, the authenticity of the PGC transcriptome data was further verified by comparing it with the transcriptome data from other cell lines obtained from the public domain [28] (Appendix A).

### 3.2. Comparison of In Vitro Cultured PGCs with In Ovo PGCs at Stage 16 HH

In this study, we cultured PGCs for a medium duration (around 70 days) in a serum-free culture medium, and we hypothesized that the in vitro media components would not have a major impact on the germ cell characteristics of PGCs. Supporting this hypothesis, these PGC lines have previously been shown to be germ-line competent [29]. To understand the influence of the cultural medium on PGCs, we first compared transcriptome profiles of cultured broiler PGCs (grown in FAOT) with in ovo PGCs (E2.5) from a layer breed of chickens [16] (Appendix A). The transcriptome profile based on the top 1000 variable genes shows a slight difference between these groups as they clustered together but were adjacent to each other (Figure 1A), and the PCA plot confirms marginally distinct transcriptome profiles between in vitro propagated and in ovo isolated PGCs (Appendix A). In addition, the biological replicates are shown to have a consistent Cook’s distance with each other, which means no outliers within each cell type (Appendix A). Based on our criteria, we identified 517 and 274 genes that are expressed higher in cultured PGCs and in ovo PGCs, respectively (Appendix A). A differential expression analysis revealed that the nucleosome organization genes (FDR: 9.89 × 10^−4^) are enriched in the cultured cells, whereas in ovo PGCs are enriched in genes involved in oxygen transport (FDR: 1.21 × 10^−4^), cellular oxidant detoxification (FDR: 4.54 × 10^−5^), proton transmembrane transport (FDR: 3.54 × 10^−5^), and signal transduction (FDR: 1.08 × 10^−2^) (Appendix A).

### 3.3. Germ Cell-Specific Gene Expression

To identify a set of genes that are exclusively expressed in chicken PGCs, we next compared the transcriptome data of chicken PGCs with the transcriptome of pluripotent cell types such as chicken embryonic stem cells (ESCs), cells from blastodermal stage embryos (EGK-X), and two differentiated somatic cells: DT40 (B cells) and primary chicken embryonic fibroblast (CEF) cells. A principal component analysis (PCA) and heatmap based on the top 1000 variable genes revealed that the biological replicates of the individual cell lines clustered together and confirmed that they had maximum similarity amongst themselves (Figure 1A and Appendix A). Similarly, the PCA revealed that the transcriptomes of different chicken PGC lines are more similar to each other in comparison to other cell lineages (Appendix A). Based on our selection criteria, a total of 242 genes were identified as germ cell-exclusive genes and are listed in (Figure 1B and Appendix A). Gene ontology enrichment analysis revealed that the piRNA metabolic process (FDR 1.20 × 10^−9^), P granule organization (FDR 1.23 × 10^−3^), spermatogenesis (FDR 9.22 × 10^−7^), DNA methylation during gamete generation (FDR 9.81 × 10^−6^), male meiotic nuclear division (FDR 8.13 × 10^−4^) (Figure 1C), and genes that are associated with the germplasm (FDR 9.10 × 10^−8^) are active in PGCs (Appendix A). The protein-protein interaction network analysis reveals germ cell-exclusive genes formed a cluster, and their protein association jointly contributes to a shared function (Figure 1D). Remarkably, the genes identified as involved in P granule organization, spermatogenesis, and DNA methylation during gamete generation were expressed at equal levels in both male and female PGCs (Appendix A). Of these, 13 protein-coding genes (*DAZL, DDX4, DDX43, MOV10L1, BMRTB1, TDRD15, FKBP6, TDRD5, GASZ, TUBA1B, ST8SIA*, and *PNLDC1*) were shortlisted based on their prior germ cell-specific expression in mammals, along with the *FDFT1, GNG10,* and *RNF17* genes as they were expressed in other cell types used for further validation (Appendix A). As a sign of validation for these genes, the candidate germ cell-specific genes were co-expressed in in vitro propagated and in ovo isolated PGCs except for the gene FDFT1 (Appendix A). In addition, the expression levels of these germ cell-specific genes were comparable between cultured and in ovo isolated PGCs (Appendix A).

### 3.4. Validation of Germ Cell-Specific Expression

To confirm the PGC-specific expression of the 16 candidate genes, we carried out an RT-qPCR analysis using cDNA samples of cultured PGCs from a layer breed for the verification of the candidate genes. Of the 16 shortlisted genes, known germ cell-specific markers such as *DAZL* and *DDX4* were used as positive controls (Appendix A). Of this set, 11 of the 14 genes were observed to be exclusively expressed in chicken PGCs (Figure 2A). To further visualize the spatial expression of the candidate genes, we performed RNA whole-mount in situ hybridization for *DAZL, DMRTB1, GTSF1, GASZ, FKBP6, PNLDC1, TUBA1B,* and the non-protein coding gene ENSGALG00010000836 (named ‘AJ’ on chicken embryos at stage 23HH, soon after PGCs colonize the gonad as demonstrated by the in situ *DAZL* RNA expression (Figure 2B). An in situ hybridization analysis revealed that *DMRTB1* and *GTSF1* were expressed solely in the putative germ cells in the gonad at this developmental stage, but *AJ, GASZ, FKBP6, PNLDC1, and TUBA1B* did not show germ cell-specific expression (Figure 2B and data not shown). Specific staining, however, was not detected elsewhere in the embryo, suggesting that the antisense RNA probes used may not be sufficiently specific for the target RNAs.

### 3.5. Identification of Differentially Expressed Genes (DEGs) between Male and Female PGCs

The transcriptome of in vitro cultured PGCs was subjected to a cluster-based analysis, which detected distinct groups for male and female PGCs (Figure 3A). One male PGC line (M19) clustered on its own but was still included in the subsequent analysis. Based on the average normalized read count (see Methods), a total of 9270 common genes were expressed in both male and female PGCs, as shown in the Venn diagram in Figure 3B. This analysis identified 82 genes that were expressed at significantly higher levels in male PGCs (Figure 3B, Appendix A). In contrast, 55 genes were expressed more highly in female PGCs (Figure 3B, Appendix A). Sex chromosome-linked genes accounted for the majority of the total number of these DEGs: 70/82 (85.4%) of male DEGs were located on the Z chromosome, and 22/55 (40%) of female DEGs were located on the W chromosome. A volcano plot reveals that the W linked genes are much more highly expressed in female PGCs, which skews the values to the left side of the axis (Figure 3C). Interestingly, a total of 55 non-protein-coding lncRNA genes are also differentially expressed between male and female PGCs. The genetic function of these lncRNAs is, however, unknown. Furthermore, a total of eight sex-specific enriched DEGs are a subset of the germ cell-specific genes identified above (Appendix A), notably *DMRT1*, *COCH,* and *FSTL1*; the remaining five genes are non-protein-coding genes (lncRNAs) (ENSGALG00000049766, ENSGALG00000026754, ENSGALG00000053408, ENSGALG00000016183 and ENSGALG00000053018).

**Figure 3 cells-12-01151-f003:**
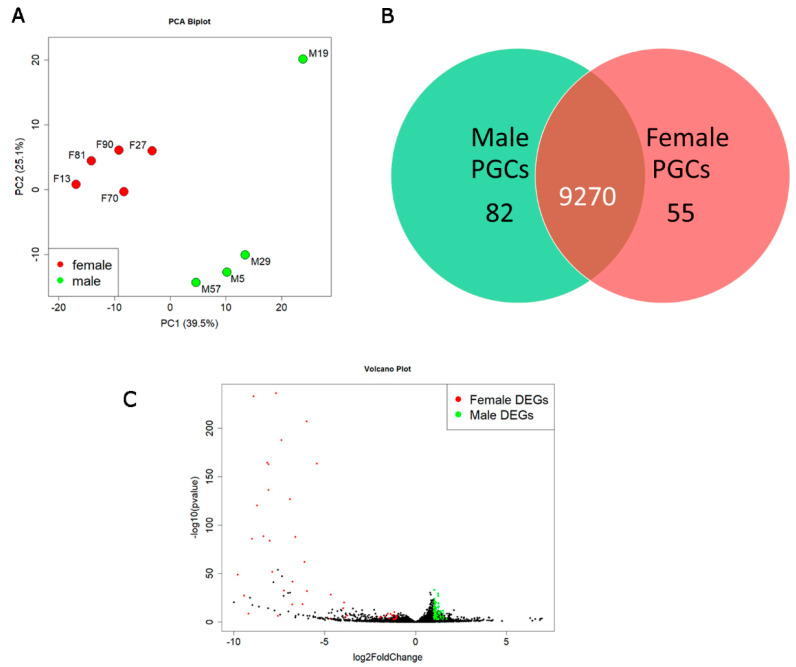
The transcriptome comparison between male and female PGCs. (**A**) PCA plot depicting the clustering of male and female chicken PGCs based on the top 1000 variable genes between male and female PGCs. (**B**) A Venn diagram showing the number of genes expressed in male and female chicken PGCs. Genes with an average normalized read count higher than 500 were considered to be expressed in a particular cell type. (**C**) The volcano plot shows the differentially expressed genes. The green and red colored dots indicate the genes that have an absolute log2fold change > 1 with an adjusted *p*-value < 0.05 and an average expression value higher than 500 in male and female PGCs, respectively (**C**).

**Figure 4 cells-12-01151-f004:**
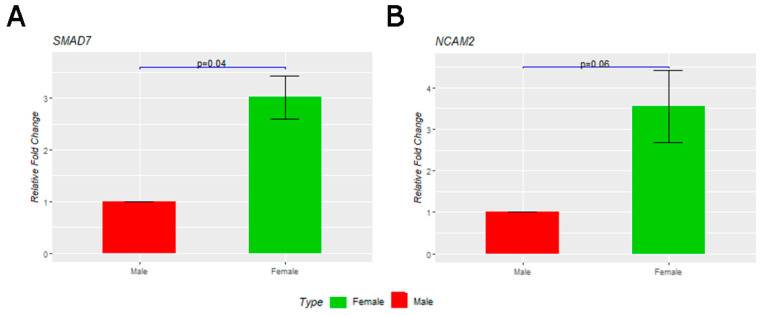
RT-qPCR validation of candidate differentially expressed genes in cultured male and female PGCs from a chicken layer breed. (**A**) The expression of the *SMAD7* gene is significantly higher in female PGCs, with a *p*-value of 0.04. (**B**) The expression of the *NCAM2* gene is higher in female PGCs than in male PGCs, with a *p*-value of 0.06.

### 3.6. Female Differentially Expressed Genes

Of the 55 genes expressed more highly in female PGCs, 22 (40.0%) are located on the W chromosome, including *WPKCI*-7, *HNRNPKL*, *UBAP2,* and *SMAD7B.* Remarkably, 21 of these genes overlap within previously identified in ovo female PGC markers [16]. A total of 18 of these 22 W-linked genes have paralogs genes on autosomes, unplaced scaffolds, or the Z chromosome. Interestingly, the paralogs expression is higher in male PGCs than in female PGCs. Three genes that are expressed more in females, ENSGALG00000041221, ENSGALG00000052041, and *HINTW*, do not have paralogous genes. In addition, *SMAD7B,* a gene in the activin signaling pathway, is expressed higher in females, and its corresponding paralogue on the Z chromosome is not expressed highly in either male or female PGCs. This infers that *SMAD7B, HINTW,* and the other two lncRNA genes are the three ‘W’ chromosome-linked genes whose expression is absent in male PGCs. An RT-PCR analysis of *SMAD7* expression showed a higher expression in female PGCs, a result which was validated by RT-qPCR analysis of both male and female PGCs, (Figure 4A). This result confirms that SMAD7 has a higher expression in female PGCs than in male PGCs (Appendix A).

The remaining 33 DEGs were located on autosomal chromosomes, with one gene located on the scaffold sequence, which is expressed higher in female PGCs. Amongst these DEGs were cell adhesion genes such as *NCAM2*, *PCDH9,* and *GRID2*, calcium ion binding genes such as *FSTL5* and *BRINP3*, the *NKD1* gene, which negatively regulates the Wnt signaling pathway, transmembrane genes *TMEM47* and *TMEM159*, and collagen-binding *COCH,* amongst others.

### 3.7. Male Differentially Expressed Genes

In male PGCs, 70 male DEGs were located on the Z chromosome (Appendix A). Notables are the genes *DMRT1, SMAD2Z, MARCH3, RPL17, XPA,* and *FBN2*. Several autosomal genes, including *NELL1*, *COL6A3*, *GFRA2*, *FAM83F*, and *RPL22L1,* are also expressed more highly in male PGCs than in female PGCs. However, none of the male DEGs overlap with the male biased genes at the E2.5 stage in ovo PGCs [16]. Of the 70 genes that are expressed higher in male PGCs than in female PGCs, 44 have paralogs on autosomal chromosomes and/or unplaced scaffolds, and their expression was comparable in both male and female PGCs. The remaining 26 genes do not have paralogs and are expressed in male PGCs but not in female PGCs. We hypothesize that somatic paralogs compensate for dosage differences in female cells.

### 3.8. Validation of Differentially Expressed Genes by Quantitative RT-PCR

The expression levels of 12 candidate male and female DEGs, associated with cell proliferation, calcium ion binding, and cell adhesion genes, were validated by RT-qPCR using independent PGC cultures from a layer breed of chicken. For female DEGs, an RT-qPCR analysis revealed that the expression of the *HINTW* gene is restricted to female PGCs, with no expression observed in male PGCs. Genes including *COCH, CRACR2B, SMAD7, DNAH3, SKAP2, TMEM159,* and *NCAM2* were expressed significantly higher in female PGCs (Figure 4A,B, Appendix A). For male-specific DEGs, *CARHSP1, FAM83F,* and *RPL22L1* were expressed significantly higher in male PGCs.

### 3.9. The Absence of Total Dosage Compensation in Chicken PGCs

As male chickens are homogametic, containing two Z chromosomes, it is expected that, on average, genes present on the Z chromosomes will be expressed at twice the level in males compared to females. However, many Z-linked genes are known to be dose-compensated in males [30]. To investigate dosage compensation in chicken PGCs for all Z-located genes, we performed a distribution comparison of expression (fold change) differences between the male and female cultured PGCs for all chromosomes. We compared Z chromosome expression levels between male and female PGCs. We also compared autosomal gene expression levels between male and female PGCs. The median of log2fold change values for the Z-linked chromosome is higher in male PGCs in comparison to the autosomal chromosomes. (Figure 5A). Surprisingly, the median value for Z-linked genes (median 0.72) is less than the expected (1.0), indicating a partial dosage compensation exists in male PGCs. As expected, the density of log2fold change of Z-linked genes is shifted to the right as compared to the density of all autosomal-linked genes (Figure 5B), due to many of the Z-linked genes having log2fold change values in the range of 1.0 to 1.5 (Appendix A). Moreover, combining in ovo and cultured PGCs resulted in a similar outcome (Appendix A), indicating this difference is not due to in vitro culturing.

## 4. Discussion

In summary, we have generated transcriptome data from the in vitro propagated chicken PGCs isolated from the circulatory stage of development and derived from a commercial broiler chicken line. A comparison of the transcriptomes of these male and female chicken PGCs, which were cultured in a serum-free culture medium, reveals that male and female PGCs are transcriptionally distinct. Notably, all the candidate germ cell-specific genes and genes involved in germ cell development are co-expressed in in vitro propagated and in ovo isolated PGCs [16]. Further characterization and functional studies of these genes will drive the prospects of generating PGC-like cells from pluripotent cells and therefore the in vitro derivation of gametes.

Our study also revealed that dosage compensation is inconsistent throughout the Z chromosome in PGCs, which supports the incomplete dosage mechanism observed in birds. There is wide evidence showing that birds do not have complete dosage compensation of the sex chromosomes [6,31,32] and it has been reported that dosage compensation is local, regulated at the gene-by-gene level, and varies with the type of tissue [7,30,33]. Our transcriptome data show that Z chromosome-linked genes have higher log2fold change values (m/f) compared to autosomes (Figure 5A). It has also been shown that dosage compensation is not equal throughout the length of the Z chromosome [30,34]. Our results support this observation, as the expression of Z chromosome-linked genes in male PGCs is not the same for genes located throughout the overall length of the Z chromosome (Appendix A).

Our results suggest that TGFβ signaling pathway genes are active, and cell adhesion genes such as *NCAM2*, *PCDH9,* and *GRID2*, and calcium ion binding genes such as *FSTL5* and *BRINP3* are expressed higher in female PGCs than in male PGCs. Overexpression of the *NCAM2* and *SMAD7* genes in female cells is one possible reason for their slower proliferation and increased cell-cell adhesion properties (Figure 6). *NCAM2* is a candidate gene that influences cell adherence in female PGCs; however, the exact functional role of the *SMAD7* gene is unknown.

The Smad family genes, *SMAD6* and *SMAD7,* are key regulators of TGFβ/BMP signaling pathways by way of a negative feedback loop [35]. The *SMAD7* gene is an antagonist to TGFβ/Activin signaling by interacting with the TGF-beta receptor type-1 and the Activin receptor, leading to inhibition of phosphorylation of Smad1/2. The overexpression of *SMAD7* promotes cell adhesion by increasing the binding of beta-catenin to E-cadherin complexes [36]. It is also shown that inhibiting the expression of *SMAD7* can prevent TGF-mediated apoptosis, signifying a role for *SMAD7* in modulating apoptosis [37]. Many studies have shown a potential therapeutic role of the SMAD7 protein for several human diseases by antagonizing TGFβ-mediated fibrosis, homeostasis, carcinogenesis, and inflammation [38].

The chicken genome has two copies of *SMAD7* that are located on each of the sex chromosomes ‘W’ and ‘Z’. The transcriptome analysis reveals that the gene residing on the Z chromosome is not expressed in either male or female PGCs, and the ‘W’ linked gene is only expressed in female PGCs. The higher expression of *SMAD7* in female PGCs suggests that the TGFβ signaling pathway is different in female PGCs (Figure 6A and 6B). Thus, expression of the *SMAD7B* gene in female cells is a possible reason for their slower proliferation and increased cell–cell adhesion properties.

PGCs have the unique characteristic of being able to achieve pluripotency under the appropriate developmental signals. The comparison of the transcriptomes of PGCs and other cell lines enables us to identify the novel germ cell-specific genes along with the well-known germ cell-specific markers. The characterization of these germ cell-specific genes may allow insight into the function of these genes in germ cell development/survival and the crucial role of these genes in germ cell biology.

## Figures and Tables

**Figure 1 cells-12-01151-f001:**
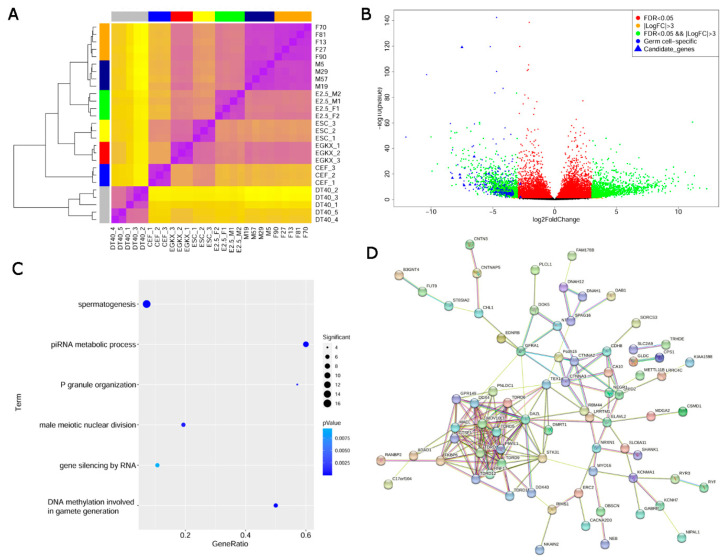
The transcriptome comparison between PGCs and other cell lines. (**A**) Heatmap showing the sample distances based on 1000 variable genes of chicken PGCs, in ovo PGCs (E2.5), chicken embryonic fibroblasts (CEF), DT40 cell line (DT40), embryonic stem cells (ESCs), and blastodermal cells (EGKX cells) from data in the public domain. (**B**) The volcano plot showing germ cell-exclusive genes. (**C**) Gene ontology of transcripts that are exclusively expressed in germ cells. (**D**) Protein-protein interaction networks in the germ cell-exclusive genes obtained from the STRING database. A tight cluster contains a set of highly connected nodes, each node represents a protein-coding gene locus; the edges represent protein-protein associations; and the colored lines represent the type of interactions. Purple: experimental evidence; green: gene neighborhood; blue: gene co-occurrence database evidence; yellow: text mining evidence; and black: co-expression evidence.

**Figure 2 cells-12-01151-f002:**
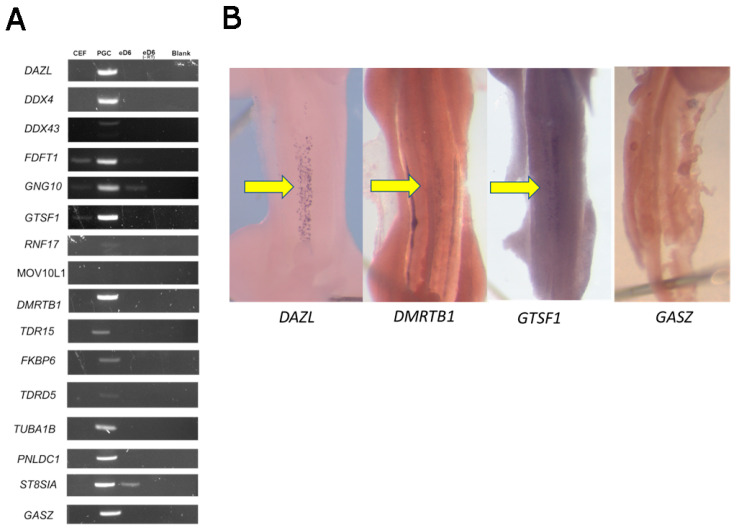
Validation of germ cell-specific expression. (**A**) An RT-PCR analysis was performed for 16 protein-coding genes in chicken embryonic fibroblasts (CEF), chicken embryos of day 6 (eD6) without mesonephroi, cultured chicken primordial germ cells (PGC), and chicken day six embryos without the reverse transcriptase enzyme eD6 (-RT). (**B**) Whole-mount RNA in-situ hybridization for candidate germ cell-specific genes on chicken embryos. A specific signal was detected for the *DAZL*, *DMRTB1,* and *GTSF1* probes, but GASZ showed no or non-PGC-specific expression. The yellow arrows indicate the genital ridges with labeled PGCs.

**Figure 5 cells-12-01151-f005:**
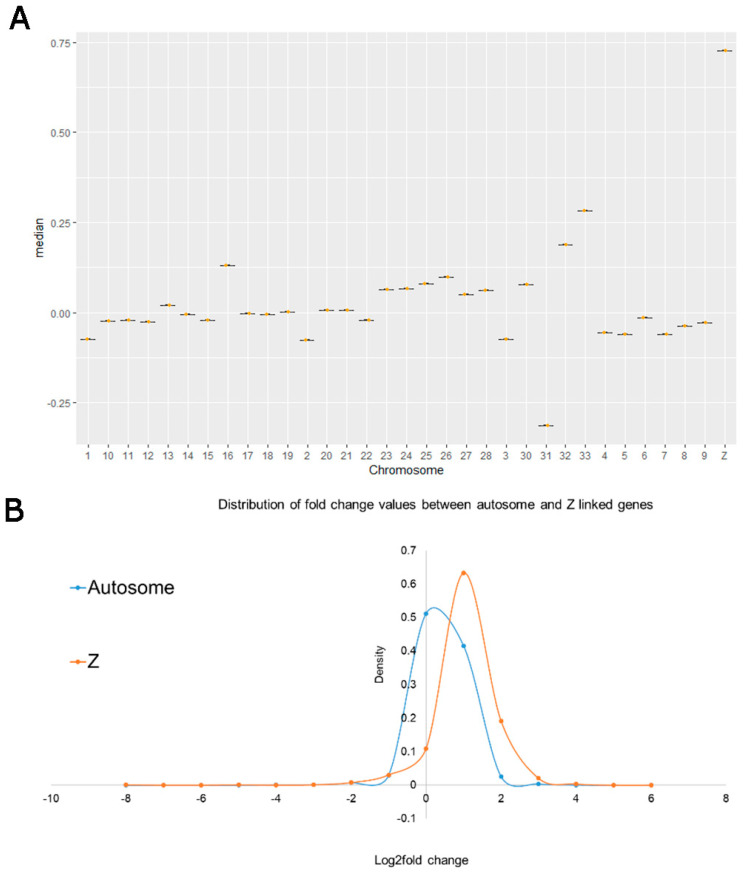
(**A**) Chromosome-wise folding changes the expression of genes in the cultured PGCs. The boxplot shows the median of log2 male to female fold change values per chromosome. The median value for the Z chromosome is the highest compared to all other chromosomes. (**B**) Comparison of the fold change distribution of autosomes and Z chromosome-linked genes. The peak density of Z-linked chromosome genes is slightly less varied than the density of autosomal genes. It shows that overall, Z chromosome-linked genes have higher log2 fold change values than autosomes.

**Figure 6 cells-12-01151-f006:**
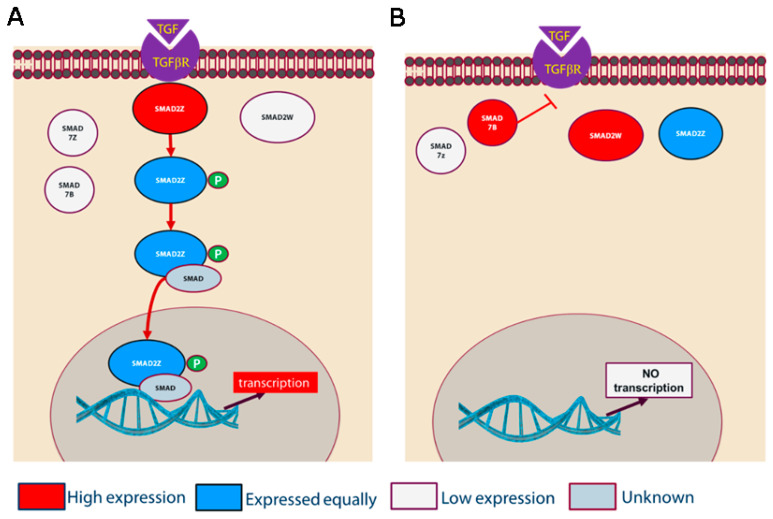
A model of the TGFβ/Activin-signaling pathway in chicken male (**A**) and female (**B**) PGCs.

## Data Availability

Chicken circulatory PGC RNA-seq data sets generated in this study are publicly available at the NCBI SRA database under accession number “PRJNA856993”.

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
