# Peer review of "The Transcriptome of Chicken Migratory Primordial Germ Cells Reveals Intrinsic Sex Differences and Expression of Hallmark Germ Cell Genes"

_cells, 2023, doi:10.3390/cells12081151_

Round 1

Reviewer 1 Report

The authors Doddamani et al., investigate and compare the transcriptomic profile between cultured primordial germ cells (PGCs), circulating ones and other chicken stem and somatic cells.

As a general comment, the manuscript is very clear, complete and the data support the results and some hypotheses. Very few comments can be made and just a few details deserve to be clarified by the authors.

In Materials and Methods, it is surprising that the different conditions for comparing transcriptomic sets are not identical: for PGC-specific genes the normalized expression is 700 with a log FC > 2, whereas it is 600 and a log2FC > 3 for the PGCs in ovo -dataset of Ischikawa et al. Please comment.

In figure 1A, a very clear separation between somatic cells and the other cell types is observed. But, even so close, the position in the heatmap of the blastoderm cells (EGKX) seems closer to the PGCs in culture in vitro than these with the PGCs in circulation (E2.5). This may appear surprising, because the contribution of PGCs to blastoderm cells is very minor (50 to 60 PGCs) as mentioned in the introduction? Could the authors comment and indicate which genes in particular are most concerned.

Figure 1D: It would be more readable to remove the non "connected" genes to better visualize the most interesting subclusters as mentioned in S4.

Figure 4 could include the data presented in S8 by separating specific male and specific female genes into two distinct parts. The expression of SMAD7 is presented, but what about the SMAD7B validation ? Additionally, the Figure 6 – not mentioned in the text- is really unclear and should just be proposed as an hypothesis as female PGCs are still proliferating, even if at an usual lower rate than the male ones ? Please modify and adapt both text and figure.

Figure 5: it is not clear if the presented data are from cultured or circulating PGCs? Does the comparison between the two cell types modify this median value?

In Discussion part, the first sentence is not explicit. The data on the transcriptome of circulating cells are indeed those of Ichikawa et al. ? Also, as previously mentioned, the analysis of TGFb signalling pathway should be more explicit.

In conclusion, by answering and modifying these few elements and remarks, the manuscript can be published.

Author Response

The authors Doddamani et al., investigate and compare the transcriptomic profile between cultured primordial germ cells (PGCs), circulating ones and other chicken stem and somatic cells.

As a general comment, the manuscript is very clear, complete and the data support the results and some hypotheses. Very few comments can be made and just a few details deserve to be clarified by the authors.

  1. In Materials and Methods, it is surprising that the different conditions for comparing transcriptomic sets are not identical: for PGC-specific genes the normalized expression is 700 with a log FC > 2, whereas it is 600 and a log2FC > 3 for the PGCs in ovo -dataset of Ischikawa et al. Please comment.

Authors' response: We apologise for the inconsistency in defining threshold values here. Now, we have made uniform threshold values of Log2FC >3 and average expression value >700 for identifying differentially expressed genes between cultured and in ovo and identifying germ cell specific genes. We made these changes throughout the revised manuscript.

  1. In figure 1A, a very clear separation between somatic cells and the other cell types is observed. But, even so close, the position in the heatmap of the blastoderm cells (EGKX) seems closer to the PGCs in culture in vitro than these with the PGCs in circulation (E2.5). This may appear surprising, because the contribution of PGCs to blastoderm cells is very minor (50 to 60 PGCs) as mentioned in the introduction? Could the authors comment and indicate which genes in particular are most concerned.

Authors' response: The difference between the heat map (Figure1A) and the PCA plot (Supplementary Figure S2A) is that we did not compare the same transcription profile. We have now compared the 1000 top variable genes for both figures and the clustering is now similar between the two figures.  

  1. Figure 1D: It would be more readable to remove the non "connected" genes to better visualize the most interesting subclusters as mentioned in S4.

Authors' response: We have updated Figure 1D by retaining only connected genes in a protein interaction network.

  1. Figure 4 could include the data presented in S8 by separating specific male and specific female genes into two distinct parts. The expression of SMAD7 is presented, but what about the SMAD7B validation ? Additionally, the Figure 6 – not mentioned in the text- is really unclear and should just be proposed as an hypothesis as female PGCs are still proliferating, even if at an usual lower rate than the male ones ? Please modify and adapt both text and figure.

Authors’ response: Thank you for pointing this out. We designed the primers in such a way that we could amplify both SMAD7Z and SMAD7B. Supplementary Fig. S7 shows amplification of SMAD7Z in male cells, SMAD7Z and SMAD7B in female cells. The female cells show higher expression (brighter band) in RT-PCR experiment. The alignment of the female and male RT-PCR products with the Z and W Smad genes is now shown in Suplementary Fig. S7B.

We apologise for not making it clear about Figure 6 in main text. Yes, we could hypothesize that TGF beta signalling pathway is affected in the female cells led to lower rate of proliferation. Now we have included citation of Figure 6 in the discussion.  (Please refer page no 16 and line number 388 in the revised manuscript.)

  1. Figure 5: it is not clear if the presented data are from cultured or circulating PGCs? Does the comparison between the two cell types modify this median value?

Authors’ response: We apologise for the lack of clarity here. The data presented in the Figure 5 is from cultured PGCs. We performed an analysis using both cultured and in ovo PGCs data, and the results remained same. We have included these results in supplementary Figure S11.

  1. In Discussion part, the first sentence is not explicit. The data on the transcriptome of circulating cells are indeed those of Ichikawa et al. ? Also, as previously mentioned, the analysis of TGFb signalling pathway should be more explicit.

Authors’ response: We have generated transcriptome data from in vitro propagated chicken isolated from the circulatory system of embryos. We have made appropriate changes in the revised manuscript and cited Ichikawa again.

Reviewer 2 Report

Summary of the study:

In this manuscript, the transcriptome of 70- to 90-day-old chicken PGCs (cPGCs) has been compared with the transcriptome of 2.5-day-old cPGCs (in ovo PGCs at stage 16 HH) and several differentiated and pluripotent cells to find exclusive and highly expressed germ-cell- and sex-specific genes which are involved in germ cell development and function.

 Overall assessment:

This is an interesting study. The findings of this study can be used for the deciphering of a more reliable culture condition for the in-vitro propagation of cPGCs and a better understanding of the biological differences between male and female PGCs in chickens.

 Specific points:

1. Title can be more clear. One suggestion is: The transcriptome of chicken migratory primordial germ cells reveals intrinsic sex differences and expression of hallmark germ cell genes.

2. The section ‘Results 3.1’ is an extension of ‘Materials and Methods’ and belongs to the section ‘2.4. Generation of RNAseq data and genome mapping’.

3. In Figure 1, it is not clear what the abbreviated names of cell lines are referring to.   

4. In Figure S2 legend, it should be mentioned that the samples are clustered based on their similarity.

5. Results 3.2: Is the sentence “… and the PCA plot based on the top 1000 variable genes confirming a distinct transcriptome between in vitro and in ovo PGCs.” correct? I see very close and similar clusters of in vitro and in ovo PGCs in Figure S2.

6. Results 3.2: The text mentions that the enrichment of the nucleosome organization genes (FDR: 9.89E-04) in the cultured cells was observed. However, this point has not been shown in Figure S3A.

7. Result 3.4: It is not clear what kind of PGCs were subjected to RNA isolation and RT-PCR. The authors need to emphasize how the RT-PCR data relate to the data obtained in situ hybridization at stage 23HH. They also need to compare the sensitivity of these techniques.

8. The ‘eD6’ and ‘CEF’ should be explained in the legend of Figure 2.

9. A better and more complete explanation of Figure 3 is needed. For example, the M19 makes a different cluster than the other male cell lines, and what the Venn diagram is showing.

10. The correct acronym is RT-qPCR and should be used instead of qRT-PCR.  

11. The method of analysis and normalization of qPCR data has not been fully described. The analysis method and the criteria to justify the analysis method (delta-delta Ct) have not been mentioned. The samples of melting curves, amplification curves, and standard curves should be included (perhaps as supplementary figures). The efficiency of the qPCR reactions has not been indicated.  No data have been presented that shows how the experiments have undergone quality controls (including the quality of RNA, the identity of PCR bands, PCR optimization experiments, etc.). The MIQE  https://rdml.org/miqe.html) requirements are not included.

12. I am having difficulty understanding the concept of comparing the expression of genes on the z-chromosome and the genes on autosomes. The concept, methodology, and calculations need much more explanation.  The comparison relative to female PGCs is not clear at all.

13. This manuscript reports that cultured male cPGCs contain distinct transcriptome features than those obtained from the E2.5 stage (in ovo cPGCs). These distinct transcriptome features may have originated from the inappropriate culture condition for male cPGCs.  

14. The ‘Abstract’ should be revised. The current version is not able to fully summarize the findings.  

Author Response

Summary of the study:

In this manuscript, the transcriptome of 70- to 90-day-old chicken PGCs (cPGCs) has been compared with the transcriptome of 2.5-day-old cPGCs (in ovo PGCs at stage 16 HH) and several differentiated and pluripotent cells to find exclusive and highly expressed germ-cell- and sex-specific genes which are involved in germ cell development and function.

 Overall assessment:

This is an interesting study. The findings of this study can be used for the deciphering of a more reliable culture condition for the in-vitro propagation of cPGCs and a better understanding of the biological differences between male and female PGCs in chickens.

Authors’ response:

We appreciate the reviewer’s constructive feedback on the manuscript. We agree with the reviewer that our study will help to improve the invitro cultural condition of chicken PGCs We have addressed your comments below and feel that the work is greatly improved as a result of your input.

Specific points:

  1. Title can be more clear. One suggestion is: The transcriptome of chicken migratory primordial germ cells reveals intrinsic sex differences and expression ofhallmark germ cell genes.

Authors' response: We thank the reviewers' for the suggestion and have adopted the new title.

  1. The section ‘Results 3.1’ is an extension of ‘Materials and Methods’ and belongs to the section ‘2.4. Generation of RNAseq data and genome mapping’.

Authors’ response: Thank you for pointing this out. We made appropriate changes in the revised manuscript.

  1. In Figure 1, it is not clear what the abbreviated names of cell lines are referring to.   

Authors’ response: Thank you for pointing this out. We have added full cell names with abbreviations in the figure legend.

  1. In Figure S2 legend, it should be mentioned that the samples are clustered based on their similarity.

Authors’ response: Thank you for pointing this out. We made appropriate changes in the revised manuscript.

  1. Results 3.2: Is the sentence “… and the PCA plot based on the top 1000 variable genes confirming a distinct transcriptome between in vitro and in ovo PGCs.” correct? I see very close and similar clusters of in vitro and in ovo PGCs in Figure S2.

Authors’ response: We observed in vitro and in ovo PGCs are in close proximity to each other and have slightly different transcriptome profile. We made appropriate changes in the revised manuscript.

  1. Results 3.2: The text mentions that the enrichment of the nucleosome organization genes (FDR: 9.89E-04) in the cultured cells was observed. However, this point has not been shown in Figure S3A.

Authors’ response: We apologise for the lack of clarity here. The genes that are expressed higher in cultured PGCs have shown significant enrichment in the nucleosome organisation bioprocess alone. Whereas, the figure S3A shows gene ontology enrichment of genes that are expressed higher in in ovo PGCs.

  1. Result 3.4: It is not clear what kind of PGCs were subjected to RNA isolation and RT-PCR. The authors need to emphasize how the RT-PCR data relate to the data obtained in situ hybridization at stage 23HH. They also need to compare the sensitivity of these techniques.

Authors’ response: We apologise for the lack of clarity here. We isolated and culture PGCs from a layer breed of chickens. RNA was analysed from these cultures. We added this information.

RNA in situ hybridisation, is sometimes as sensitive as RT-PCR. However, this is very probe specific. Many RNAs that are highly expressed by RT-PCR analysis and RNAseq may not give a spatial signal by RNA in situ hybridisation. We have added a statement to this section.

  1. The ‘eD6’ and ‘CEF’ should be explained in the legend of Figure 2.

Authors’ response: Thank you for pointing this out. We made appropriate changes in the revised manuscript.

  1. A better and more complete explanation of Figure 3 is needed. For example, the M19 makes a different cluster than the other male cell lines, and what the Venn diagram is showing.

Authors’ response: We have now commented on the outlier male PGC line, all we can add is that this line is an outlier but we used it in our analysis. We also added more description to page 7 explaining the Venn diagram and the Volcano plot.

  1. The correct acronym is RT-qPCR and should be used instead of qRT-PCR. 

Authors’ response: Thank you for pointing this out. We made appropriate changes in the revised manuscript.

  1. The method of analysis and normalization of qPCR data has not been fully described. The analysis method and the criteria to justify the analysis method (delta-delta Ct) have not been mentioned. The samples of melting curves, amplification curves, and standard curves should be included (perhaps as supplementary figures). The efficiency of the qPCR reactions has not been indicated.  No data have been presented that shows how the experiments have undergone quality controls (including the quality of RNA, the identity of PCR bands, PCR optimization experiments, etc.). The MIQE  https://rdml.org/miqe.html) requirements are not included.

We have added more information to the materials and methods and added the melting curves in the Supplementary Data section.

  1. I am having difficulty understanding the concept of comparing the expression of genes on the z-chromosome and the genes on autosomes. The concept, methodology, and calculations need much more explanation.  The comparison relative to female PGCs is not clear at all.

Authors’ response: We apologise for the lack of clarity here. We have re-written that section to make the analysis clearer to the reader.

  1. This manuscript reports that cultured male cPGCs contain distinct transcriptome features than those obtained from the E2.5 stage (in ovo cPGCs). These distinct transcriptome features may have originated from the inappropriate culture condition for male cPGCs.  

Authors’ response: We have revised this paragraph (see reviewer 1), The cultured and in ovo PGCs are similar but with slight differences which may be due to culturing or to the differences in the chicken breeds used in the two analyses. This was pointed out in the text.

Male and female PGCs cultured using this culture medium are germ line competent (Whyte, Stem cell reports, 2017; Woodcock, PNAS, 2019; Ballantyne, Nature communications, 2021). so we do conclude that the medium is appropriate for both male and female germ cells.

  1. The ‘Abstract’ should be revised. The current version is not able to fully summarize the findings.  

Authors’ response: We have revised the abstract.

Round 2

Reviewer 2 Report

The revised version of the manuscript is much improved, and my comments have been satisfactorily addressed.